# One-Piece Zirconia Oral Implants for the Support of Three-Unit Fixed Dental Prostheses: Three-Year Results from a Prospective Case Series

**DOI:** 10.3390/jfb14010045

**Published:** 2023-01-13

**Authors:** Ralf-Joachim Kohal, Kirstin Vach, Frank Butz, Benedikt Christopher Spies, Sebastian Berthold Maximilian Patzelt, Felix Burkhardt

**Affiliations:** 1Department of Prosthetic Dentistry, Faculty of Medicine, Center for Dental Medicine, Medical Center-University of Freiburg, University of Freiburg, Hugstetter Str. 55, 79106 Freiburg, Germany; 2Institute of Medical Biometry and Statistics, Faculty of Medicine, Medical Center-University of Freiburg, University of Freiburg, Zinkmattenstr. 6a, 79108 Freiburg, Germany; 3Private Dental Clinic, Belchenstr. 6a, 79189 Bad Krozingen, Germany; 4Private Dental Clinic, Am Dorfplatz 3, 78658 Zimmern ob Rottweil, Germany

**Keywords:** clinical investigation, oral implant, fixed dental prosthesis, zirconia implant, prospective

## Abstract

The objective was to investigate the clinical and radiological outcome of one-piece zirconia oral implants to support three-unit fixed dental prostheses (FDP) after three years in function. Twenty-seven patients were treated with a total of 54 implants in a one-stage surgery and immediate provisionalization. Standardized radiographs were taken at implant placement, after one year and after three years, to evaluate peri-implant bone loss. Soft-tissue parameters were also assessed. Linear mixed regression models as well as Wilcoxon Signed Rank tests were used for analyzing differences between groups and time points (*p* < 0.05). At the three-year evaluation, one implant was lost, resulting in a cumulative survival rate of 98.1%. The mean marginal bone loss amounted to 2.16 mm. An implant success grade I of 52% (bone loss of ≤2 mm) and success grade II of 61% (bone loss of ≤3 mm) were achieved. None of the evaluated baseline parameters affected bone loss. The survival rate of the zirconia implants was comparable to market-available titanium implants. However, an increased marginal bone loss was observed with a high peri-implantitis incidence and a resulting low implant success rate.

## 1. Introduction

Based on their reliable osseointegration and long-term clinical success, titanium implants with microroughened surfaces are still the gold standard in oral implantology [1,2]. However, patients’ demands for metal-free alternatives increase as titanium particles in the peri-implant soft tissue might be potentially hazardous to health [3,4]. 

Due to computer-assisted manufacturing, zirconium dioxide (zirconia, ZrO_2_), mostly used as yttria-stabilized tetragonal zirconia polycrystal (3Y-TZP), increasingly gained importance as a dental restoration material [5]. Compared to other ceramics, ZrO_2_ exhibits metal-like biomechanical properties, which are reflected in superior bending strength and fracture toughness [6]. This can also be attributed to the mechanism of phase transformation toughening, which leads to compression of the matrix and thus counteracts crack growth [7]. Furthermore, plaque shows a low affinity to ZrO_2_ [8] and seems to be soft tissue friendly due to low inflammation levels when applied as an oral implant material [9]. The tooth-like color of the ceramic can avoid the visual appearance of grayish titanium in the case of a thin gingival biotype [10]. 

The first one-piece zirconia oral implants were introduced to the market in the early 2000s [11]. Similar to titanium, zirconia implants with a micro-roughened surface showed increased bone-to-implant contact and, consequently, a comparable osseointegration to titanium implants [12,13]. Initially, the challenge was to obtain an implant design with a microrough surface without reducing the stability of the zirconia implants. Surface modifications can increase the risk of aging of the ZrO_2_ in the warm and humid environment of the oral cavity resulting in a reduced fracture resistance of the implants [14,15]. Nevertheless, zirconia implants have been significantly improved in recent years through optimization of geometry and surface treatments, thus making one-piece zirconia implants a valid treatment option as demonstrated in preclinical [16,17] and clinical studies [18]. 

Nowadays, common methods for the manufacturing of microrough surfaces of the endosseous implant parts are subtractive methods such as sandblasting, acid etching or laser treatment [15]. As an alternative, there are additive methods to achieve porous surfaces, e.g., by coating the implants with a zirconia powder [19] and a pore former before the final sintering. The roughened surface of the investigated implant was achieved by coating the implants with slurry containing zirconia powder and a pore former (patent SE0302539-2) [20]. The implants were subsequently sintered to full density. This resulted in the pore former burning out completely and leaving a porous surface [21].

In animal studies, implants with surfaces treated with an additive method exhibited four to five times higher resistance in removal torque tests after a healing period of six weeks in the rabbit bone than zirconia implants with machined surfaces [21,22].

Although the number of prospective clinical trials on one-piece zirconia implants has increased in recent years, investigations of ceramic implants as abutments for fixed dental prostheses (FDP) are scarce [18,23]. Therefore, the objective of this prospective long-term cohort study was the clinical and radiological evaluation of a one-piece zirconia implant system with additively coated surfaces for the support of three-unit FDP after three years of implant placement. The null hypothesis of this prospective case series was that (a) the survival and (b) the success rate of the applied zirconia implant system is not different than titanium oral implants after 3 years.

## 2. Materials and Methods

### 2.1. Study Population 

All of the 27 patients participating in this prospective cohort study agreed and signed an informed consent before treatment. Inclusion and exclusion criteria were previously described in detail [24]. The study protocol was in compliance with the STROBE statement for observational cohort studies and approved by the local ethics committee (investigation number: 337/04; University Clinics Freiburg, Freiburg, Germany).

### 2.2. Clinical Assessment 

#### 2.2.1. Investigated Implant System

The investigated one-piece oral implants were made of 3Y-TZP (ZiUnite, Nobel Biocare, Gothenburg, Sweden). The implant design was similar to the Nobel Direct titanium implant (Nobel Biocare) with a threaded tapered endosseous part, a transmucosal cylindrical collar and an integrated abutment cylinder (Figure 1). The endosseous part and transmucosal collar were moderately roughened (arithmetic mean height (S_a_) = 1.24 μm; density of summits (S_ds_) = 0.09/μm^2^; Surface Area Ratio (S_dr_) = 82.6%) with the ZiUnite surface technology (Nobel Biocare) leading to a porous surface achieved by sintering the implants with a zirconia slurry and burnable pore former [21] (Figure 2). The surface of the abutment part had a machined surface. The implants were available in 10, 13 and 16 mm lengths, and with regular (RP Ø 4.3 mm) and wide platform diameters (WP Ø 5.0 mm). The implants were CE-marked but classified as investigational by the manufacturer. The implants were never placed on the market. 

#### 2.2.2. Implant Surgery and Prosthetic Delivery

A radiographic examination of the prospective implant area was performed using cone beam computed tomography (Newtom 3G, Newtom, Marburg, Germany) to select the appropriate length and diameter of the implants. Both, the surgical and prosthetic procedures, were previously described in detail [24]. Implant placement was either performed immediately after extraction or after a healing period of at least four to six months. Bone quality and quantity were clinically assessed during the surgery according to Lekholm and Zarb [25]. After implant placement, the abutment part was slightly prepared for seating of a polymethyl methacrylate (PMMA) temporary three-unit prosthesis. Occlusal and lateral contacts of the temporary prosthesis were removed to an infra-occlusal level to avoid load application during healing. Patients were instructed to avoid load in the area of the implants, but no further interventions were undertaken to reduce load (e.g., protective splints, dietary restrictions). After two or four months after implant placement, the final FDP were delivered for the mandible and maxilla, respectively. The three-unit FDP consisted of a zirconia framework (Procera) and a glass-ceramic veneering (NobelRondo, both Nobel Biocare) which was commonly used at the time when the study was initiated. All restorations were cemented with a glass ionomer cement (Ketac Cem, 3M Espe, Neuss, Germany), and cement remnants were meticulously removed.

#### 2.2.3. Follow-Ups

Standardized radiographs were taken using the parallel technique at the day of implant placement (baseline), at the day of FDP delivery, at the 1-year and the 3-year follow-ups and independently investigated by a radiologist at the University of Gothenburg, Sweden. Bone remodeling was measured by calibrating the radiographs using the known width of the base of the implant abutment part. The lower edge of the abutment part served as reference for the measurements (Figure 1). The reference points differed in height between the mesial and distal implant region based on the different start of the threads. Therefore, the bone remodeling was measured mesially and distally, and a mean value was calculated. Clinical examinations of the implants as well as the adjacent teeth, which served as reference teeth, included probing depths (PD), clinical attachment levels (CAL), bleeding on probing (BOP) and plaque index (PI). The clinical examinations were performed at the prosthesis delivery and at the 1-year and 3-year follow-ups, and patients were included in an annual recall program. In addition, impressions were taken for study casts, and changes in medical history were recorded. The implant success criteria of Östman, Hellman, Albrektsson & Sennerby [26] were utilized in a slight modification, with success grade I applied to implants with a bone resorption of ≤2 mm and success grade II to implants with a bone resorption of ≤3 mm. 

### 2.3. Statistical Analysis 

Mean values and standard deviations are reported for normally distributed data; otherwise, the median and the 25% or 75% quartiles are given. The cumulative survival rates of the zirconia implants were calculated by means of an actuarial life table analysis [27]. The missing bone level values at implant placement were imputed by using the information on the following time point and the mean changes of the whole population. For analyzing changes as well as subgroup differences in bone levels, PD and CAL linear mixed models were used; in subsequent pairwise comparisons, the Scheffé method was used to correct for multiple testing. As the values for BOP and plaque index were not normally distributed, Wilcoxon Signed Rank tests on patient level was applied to test for differences between teeth and implants. The level of significance was set to 0.05. All calculations were performed with the statistical software STATA 17.0 (StataCorp, College Station, TX, USA).

## 3. Results

A total of 54 implants were placed in 27 patients (11 females/17 males) to support 27 three-unit FDP. The age distribution of the patients at the implant surgeries was as follows: One patient was between 18 and 30, and eight patients were between 31 and 50 years old. Nineteen patients were in the age group of 51 to 70 years. All patients were non-smokers. Forty-four implants were inserted in the lower jaws (all in the posterior area) and 10 implants in the upper jaws (four in the anterior area, six in the posterior area) (Table 1). Five implants were placed in extraction sites and 49 implants in healed sites (for five implants a punch approach was performed, and two implants were inserted with a flapless technique). Most implant sites demonstrated a bone quantity of B and bone quality of II according to Lekholm and Zarb [25] (Table 2). Eleven implants exhibited insertion torques below 35 Ncm, 28 implants between 35 and 45 Ncm and 15 implants over 45 Ncm. 

### 3.1. Implant Survival 

The final FDP was delivered to 26 of the 27 patients, since one implant in one patient was lost during the healing period (Table 1). The failed implant was placed into an extraction socket and had to be removed after two weeks due to loss of stability. However, no signs of acute infection were observed. The patient was subsequently withdrawn from the study resulting in 26 patients for further evaluation. At the 1-year follow-up, all 26 patients were available for the clinical and radiographical examinations. Between the 1-year and the 3-year follow-up, one patient dropped out since he moved and was no longer available, and one patient could not attend the 3-year follow-up, leading to 24 patients available at the 3 year-follow-up. Since no further implant loss occurred, the cumulative survival rate at the 3-year follow-up was 98.1%.

### 3.2. Marginal Bone Loss 

At the 3-year follow-up, standardized radiographs of 44 implants could be analyzed and compared to baseline. From baseline to the 3-year follow-up, the mean marginal bone loss amounted to 2.16 ± 2.46 mm (Table 3, Figure 3 and Figure 4). In comparison, at the 1-year follow-up, the mean bone loss compared to baseline was 1.98 ± 2.18 mm. Three years after implant placement, 21 of the 44 implants showed a marginal bone loss of more than 2 mm (47%), while 17 implants exhibited more than 3 mm bone loss (39%). According to the success criteria of Östman et al. [26], a 3-year success grade I was achieved in 53% and success grade II in 63%. At the 1-year follow-up, 61% of the implants were successful at grade I and 63% at grade II. A univariate analysis did not show any subgroup differences neither in bone loss for the 1-year nor the 3-year follow up (Table 4). 

### 3.3. Clinical Parameters 

After delivery of the three-unit FDP, PD for both implants and reference teeth initially decreased until the 1-year follow-up (implants: 2.80 ± 0.83 mm to 2.64 ± 0.71 mm; teeth: 2.39 ± 0.70 mm to 2.02 ± 0.41 mm; Figure 5). The PD at the 3-year follow-up for the reference teeth (2.08 ± 0.63 mm) remained at the level of the 1-year follow-up and were significantly lower than at the FDP delivery (*p* = 0.042). The PD at the implant sites increased to the 3-year follow up (3.35 ± 1.74 mm) and were significantly higher compared to the PD around the reference teeth (*p* < 0.001). 

The CAL around the implants increased slightly until the 1-year follow-up (3.17 ± 0.85 mm to 3.24 ± 0.80 mm), while it decreased around the reference teeth over the same time period (3.00 ± 1.24 mm to 2.75 ± 0.81 mm; both *p* > 0.05; Figure 5). Up to the 3-year follow-up, the CAL significantly increased at the implant sites compared to the delivery of the FDP (3.92 ± 1.96 mm; *p* = 0.002), however, no statistically significant differences were observed for the reference teeth (2.59 ± 0.86 mm; *p* = 0.071). At FDP delivery, no statistically significant difference in CAL was found between implants and teeth (*p* = 0.656), whereas the differences after one year and three years (both *p* < 0.001) were statistically significant.

The BOP at implants and reference teeth decreased until the 1-year follow-up (implants: 33.82 ± 39.63%; median 25% to 16.83 ± 23.07%; median 0%; *p* = 0.043; teeth: 30.00 ± 35.25%; median 25% to 25.69 ± 28.96%; median 25%; *p* = 0.627; Figure 6). The BOP of the teeth further decreased until the 3-year follow-up (13.28 ± 22.88%; median 0%; *p* = 0.124 in relation to the FDP delivery), whereas the BOP at implant sites increased during this period (43.75 ± 39.44%; median 50; *p* = 0.124 in relation to the FDP delivery). A statistically significant difference of the BOP between implants and reference teeth was observed at the 1- and 3-year follow-up (*p* = 0.042 and *p* < 0.001).

The PI decreased significantly after FDP delivery to the 1-year follow-up for the implants (implants: 46.57 ± 44.73%; median 50% to 9.13 ± 17.87%; median 0%; *p* < 0.001) but not for the teeth (45 ± 43.22%; median 25% to 25.69 ± 35.60%; median 0%; *p* = 0.138; Figure 6). However, from the 1-year to the 3-year follow-up, both the implants and teeth showed an increase in plaque accumulation. However, it was still lower than at the FDP delivery (implants: 18.23 ± 22.91%; median 0%; teeth: 23.44 ± 31.07%; median 0%). A statistically significant difference between the implants and the reference teeth was only found at the 1-year follow-up (*p* = 0.047).

### 3.4. Biological Complications

Sixteen implants revealed a bone loss of more than 3 mm after three years. Ten of those implants showed at least a mean PD of 6 mm and suppuration/pus with the diagnosis of peri-implantitis (incidence: 23%). Peri-implant diseases were treated following the C.I.S.T. protocol [28].

## 4. Discussion

The present prospective case series describes the 3-year results of one-piece zirconia implants with a moderately rough surface for reconstruction with three-unit FDP. Key findings of the present study are an acceptable implant survival rate after three years with only one early implant loss during the healing period before prosthetic rehabilitation resulting in a cumulative survival rate of 98.1%. However, a large number of implants showed an increased marginal bone loss with a remarkable peri-implantitis incidence (23%) resulting in a low implant success rate. The null hypothesis was rejected since the success rate of the zirconia implants was reduced compared to titanium oral implants after 3 years. Although the implants were never brought to the market, the value of this study lies in the scientific observation of the subsequent mid- and long-term outcomes. In particular, the further evaluation of the initial increase in bone loss after one year seems to be of scientific and patients’ importance.

The recent growth of published clinical studies on zirconia implants reflects the increasing interest in metal-free alternatives to titanium oral implants. A systematic review and meta-analysis investigated the survival rate and marginal bone loss of zirconia oral implants over a period of 12 to 60 months [18]. Seven prospective and two randomized clinical trials could be included in this meta-analysis. Besides the 1-year report of this study on ceramic implants restored with FDP [24], only two other studies examining ceramic implants with FDP [29,30] could be included. According to the review of Pieralli et al. [18], the mean implant survival rate after one year was 95.6% for implants with single crowns (SC) and FDP. From the data of the same review, no difference in implant survival was observed between SC and FDP loaded implants. Furthermore, the review indicated that there is no statistically significant difference regarding MBL between SC (0.80 mm) and FDP (0.76 mm). Clinical trials published more recently revealed similar high survival rates [31,32,33] for implants restored with SC or FDP after at least 5 years. When comparing the applied immediately provisionalized/loaded zirconia implants to immediately loaded titanium implants of the same design (Nobel Direct), the survival rates were within the same range (94.8%) after four years [26].

Despite the high implant survival rates of the zirconia implants in the present study, the radiological analyses revealed an increased marginal bone loss, which amounted to 1.98 mm at the 1-year follow-up [24] and to 2.16 mm at the 3-year follow-up. For the short-term data of zirconia implants regarding bone loss, these values were higher than those of other zirconia implant investigations reporting a bone loss ≤ 1 mm after ≤5 years [18,23,34]. The reason for the increased bone loss in the present investigation could not be related to any of the investigated baseline parameters when applying a univariate analysis. Therefore, the reasons for the extended marginal bone loss can only be speculated upon.

In general, a moderate marginal bone loss during the first year of function of oral implants is not a pathological sign and represents normal bone remodeling in response to the surgery [35]. A progressive early bone loss due to so called aseptic loosening is often related to biological and technical factors, and potential complications instead of a disease [35]. However, some implants with initial bone loss may develop an infection of the bone-implant interface leading to peri-implantitis [36]. In the case of the Nobel Direct titanium implant, which resembles the evaluated one-piece zirconia implant, an increased initial bone loss was observed [26,37]. It was assumed that the extensive marginal bone loss was caused by the vibrations of the in situ preparation of the one-piece implants leading to an adverse bone reaction. Furthermore, implant debris produced during implant insertion or during the preparation of the abutment part of one-piece implants with the diamond bur might have led to the observed adverse bone reactions. Immediate loading may have enhanced this effect on the bone, as less crestal bone loss was observed at implants loaded after a healing phase [38]. This might have contributed to the increased initial bone loss of the evaluated zirconia implants, which were prepared with a red contra-angle handpiece (40,000 rpm) and water-cooling after placement and immediately restored with temporary prosthesis. However, the meta-analysis showed that for other types of one-piece zirconia implants, immediate temporization had no significant effect on marginal bone loss and implant failure rate [18]. This leads to the speculation that the tapered design of the implants, similarly to the Nobel Direct titanium implants, may have contributed to the increased bone loss since several investigations with this implant design have shown an increased bone loss [26,37]. The design might have exerted an increased pressure onto the crestal bone while the implants have been placed, subsequently leading to bone resorptions. Although the univariate analysis did not show any statistically significant subgroup differences, a difference of 38% of bone loss between the insertion torques of > 45 Ncm and ≤45 Ncm exists after 3 years. This could lead to the assumption that the insertion torque might have contributed to the marginal bone loss after this time period. Possibly, the post-insertion integrity of the porous and brittle implant might have suffered, and the result is being visible just after a certain time period.

Besides the radiographic bone loss, “clinical signs of inflammation, bleeding on probing and/or suppuration, and increased probing depth” indicate the presence and severity of peri-implant diseases [39]. The progressive marginal bone loss in combination with pathological clinical signs (increased PD, suppuration/pus) led to an increased incidence of peri-implantitis at the 3-year follow-up [36]. The mean PD of 3.65 mm at the implant sites at the 3-year follow-up in combination with the standard deviation (± 2.04 mm) already indicates that there are sites with increased PD although long-term clinical studies have shown that an increased PD of > 4 mm can regularly be observed at the healthy peri-implant mucosa [40,41]. In our investigation, ten of the 16 implants revealing a bone loss of more than 3 mm showed at least a mean PD of 6 mm and suppuration/pus leading to the diagnosis of peri-implantitis. The significantly increased BOP (43%) compared to the natural reference teeth (13%) represents another sign in combination with the increased PD for a high incidence of peri-implantitis at the 3-year follow-up [36,39,42]. This leads to the observed increased hard and soft tissue deficiencies at the implant sites [43]. Since peri-implant diseases are plaque associated [39], the PI was evaluated in the present study. At the 3-year follow-up, an increased PI was observed compared to the 1-year follow-up, which, however, was lower than at prosthesis insertion. Therefore, further factors besides a poor plaque control need to be considered for the increased incidence of peri-implantitis. Due to the one-piece implant design, the provisional and final restorations had to be cemented. In addition, possibly undetected cement remnants may have led to peri-implant inflammation with subsequent peri-implant bone resorption [44]. Whether this has been a reason for peri-implant bone loss in the present investigation might be questioned since systematic reviews could not identify an investigation supporting this assumption [18,23]. In addition, no cement remnants were clinically detected during follow-up examinations. After cementation, the cement remnants were thoroughly removed with magnifying glasses and checked on the radiograph.

As with titanium implants, a micro-rough surface of zirconia implants leads to an increased bone-to-implant contact and to a superior osseointegration [45]. However, the influence of surface roughness on soft tissue integration and marginal bone loss at zirconia implant sites is not yet known. However, an exposure of the micro-rough surface due to (soft or hard) tissue loss might have led to an increased bacterial colonization and subsequent infection of the crestal bone-implant interface [46,47]. An increased biofilm formation of initial adherent bacteria was observed when examining this porous ZiUnite surface in situ, which strongly correlated with the surface roughness and porosity [48]. In addition, the rough surface protects the bacteria from shear forces, which would otherwise lead to their desorption from a surface [49]. In this context, the vertical positioning of the transition zone between the micro-rough and machined surface may also have an influence on microbiological colonization, as a different extent of the micro-rough surface is exposed. However, one-piece titanium implants in which this transition was placed below the bone crest showed a higher degree of remodeling (bone loss) in the first six months than implants in which the transition zone was placed above the bone level [50].

A further reason for the increased bone loss might also be related to the surface porosity of the implants. An interconnection of the pores and thus an access of liquid to the bulk material was described in vitro for the investigated zirconia implants [51]. The penetration of water molecules could therefore have accelerated aging of the zirconia surface as well as bulk material in the warm and humid environment (low thermal degradation). After 20 hours of autoclaving at 134 °C, which roughly equates 40 years of in vivo aging, a transformation layer of 2-3 µm at the surface of the bulk material could be observed [51]. This transformation of the zirconia grains from tetragonal to monoclinic is accompanied by a volume expansion [7], which potentially causes stress and thus adverse effects on the bone. In this context, it can also be discussed that the zirconia tetragonal to monoclinic transformation at the bone-implant interface may as well be associated with particle release and subsequent “aseptic loosening”. Nevertheless, it is questionable to what extent aging might have occurred after the comparatively short periods of the present study and whether the subsequent effect of volume expansion is sufficient for bone resorption.

In summary, it may be assumed that the observed bone loss might be a combination of the various factors discussed above. A limitation of the present study is the absence of a control group. Including a second one-piece zirconia implant of a different design [31,32] could have shown whether the design has had an influence on bone remodeling. The low number of patients and the inhomogeneous cohort is another limiting factor. Due to the small subgroups, the univariate analyses can only be taken as an orientation, and analyses with larger groups are necessary. The inclusion of the patients in an (bi)annual maintenance program might have led to a decreased incidence of peri-implantitis [52].

The present study shows that the deterioration of clinical parameters obviously lags behind an increasing bone loss with the evaluated implant system. The reasons for the increased bone loss seem to be manifold. However, they finally led to peri-implant disease in the form of peri-implantitis. 

## 5. Conclusions

Despite one early implant loss during the healing period, all prosthetically restored implants were in situ after three years. This resulted in an acceptable implant survival rate, which was comparable to other (one- or two-piece) implants made of zirconia or titanium. However, bone-remodeling analyses showed a frequent occurrence of bone loss > 3 mm, which reduced the implant success rate tremendously. The bone loss after 3 years was accompanied by clinical signs of pathology with the occurrence of peri-implantitis, significantly reducing the prognosis of the implants.

## Figures and Tables

**Figure 1 jfb-14-00045-f001:**
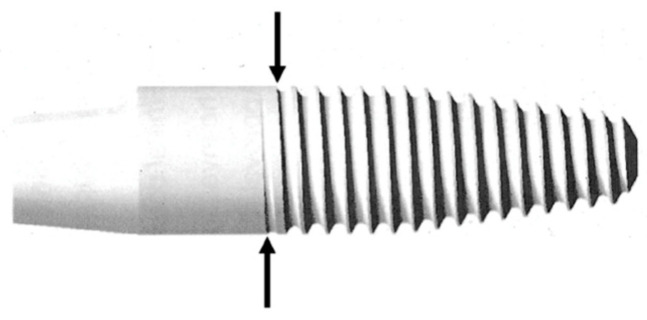
Schematic illustration of the evaluated one-piece zirconia implant (ZiUnite, Nobel Biocare, Gothenburg, Sweden). Arrows indicate the mesial and distal reference points for the radiographic evaluation of the bone level at the transition zone between the straight abutment part and the implant threads.

**Figure 2 jfb-14-00045-f002:**
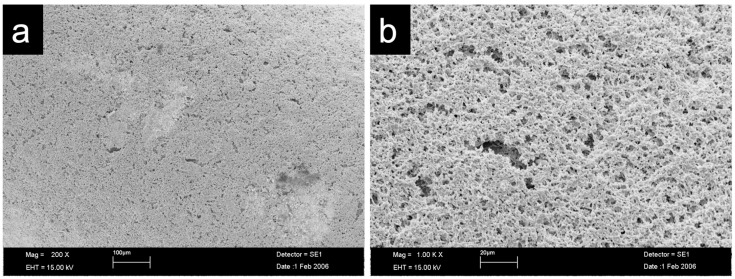
Scanning electron micrographs of the micro-roughened surface of the zirconia implant. Magnification: 200× (**a**); 1000× (**b**). Samples were critically point dried and mounted on specimen stubs and finally sputter-coated with gold palladium in the SCD 040 (Balzers Union, Wallruf, Germany). Samples were examined with the Zeiss Leo 32 scanning electron microscope (Zeiss, Oberkochen, Germany) which operated at 10–15 kV. Micropits were visible at the surface which were a result of the surface treatment described in materials and methods. The surface showed an irregular particle network with elevations and undercuts resembling a scaffold with irregular sized pores.

**Figure 3 jfb-14-00045-f003:**
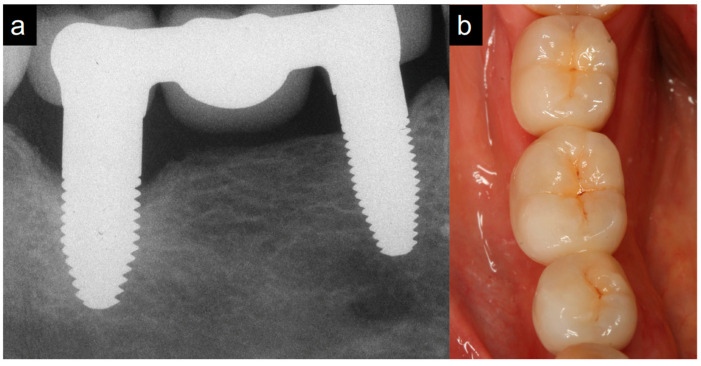
(**a**) Exemplary radiograph at the 3-year follow-up visualizing a three-unit FDP supported by two one-piece zirconia implants in the mandible. Vertical defects of the marginal bone can be observed at the distal implant. (**b**) Clinical photography of the same FDP at the 3-year follow-up.

**Figure 4 jfb-14-00045-f004:**
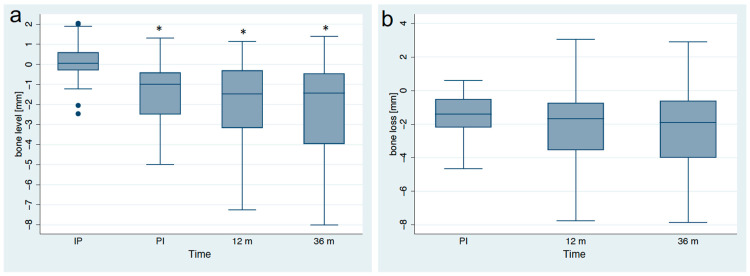
(**a**) Boxplots visualizing the bone levels (in mm) at implant placement (IP), prosthesis insertion (PI), at the 1-year (12 m) and 3-year follow-up (36 m). Asteriks (*) indicate significances (*p* < 0.001) compared to IP. (**b**) Boxplots visualizing the bone loss (in mm) in comparison to IP.

**Figure 5 jfb-14-00045-f005:**
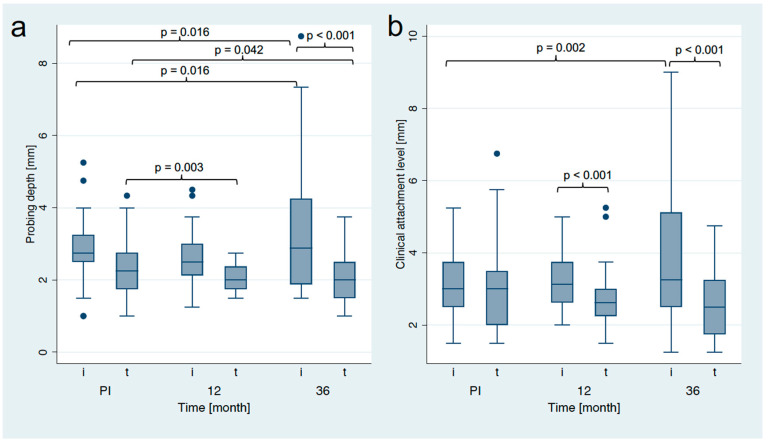
Boxplots visualizing probing depth (**a**) and clinical attachment level (**b**) at the prostheses insertion (PI), the 1-year follow-up (12 m) and the 3-year follow-up (36 m).

**Figure 6 jfb-14-00045-f006:**
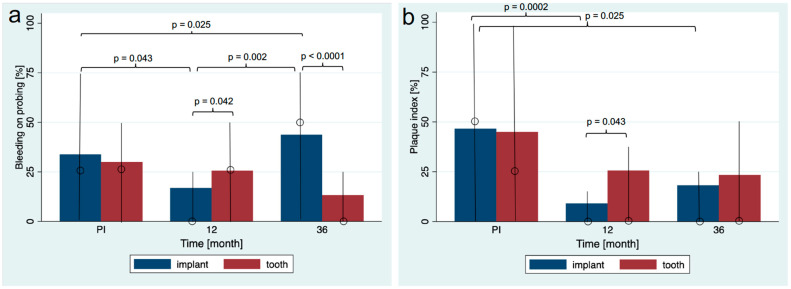
Bleeding on probing (**a**) and plaque index (**b**) at the prostheses insertion (PI), the 1-year follow-up (12 m) and the 3-year follow-up (36 m). Bars indicate the mean values and vertical lines the quartile (p25 and p75). The median is marked with a circle.

**Table 1 jfb-14-00045-t001:** Implant dimensions.

		Maxilla		Mandible	
Platform [Ø mm]	Length [mm]	Placed	Failed	Placed	Failed
4.3	10	0	0	10	0
	13	4	0	10	0
	16	0	0	1	0
	Total	4	0	21	0
5.0	10	1	0	12	1
	13	3	0	9	0
	16	0	0	2	0
	Total	4	0	23	1

**Table 2 jfb-14-00045-t002:** Bone quality and bone quantity according to Lekholm and Zarb [25].

		Bone Quality				
		1	2	3	4	Total
**Bone Quantity**	A	7	15	-	-	22
	B	2	23	-	-	25
	C	-	3	-	-	3
	D	-	2	-	-	2
	E	-	-	-	-	-
	Total	9	43	-	-	52

**Table 3 jfb-14-00045-t003:** Bone level changes according to radiographic assessments of all measurable standardized radiographs (44 at the 3-year follow-up). Negative data numbers indicate bone levels apical to the reference points (Figure 1). Bone level changes are visualized in Figure 4b.

	Implant Placement to Prosthesis Insertion	Implant Placement to Prosthesis Insertion-Imputed	Implant Placement to 1-Year Follow-Up	Implant Placement to 3-Year Follow-Up
Number	46	50	52	44
Mean Value	−1.54	−1.54	−2.09	−2.37
SD	1.32	1.26	2.04	2.42

**Table 4 jfb-14-00045-t004:** Distribution of marginal bone for 1- and 3-year follow-up in different subgroups.

	1-Year Follow-Up		3-Year Follow-Up	
	Implants (N)	Mean (SD)	Implants (N)	Mean (SD)
**Jaw Type**				
Maxilla	8	−3.16 (2.24)	5	−2.67 (2.18)
Mandible	44	−1.99 (1.97)	39	−2.33 (2.47)
**Ant-post**				
Anterior	2	−4.12 (5.13)	2	−1.03 (0.18)
Posterior	50	−2.01 (1.91)	42	−2.46 (2.46)
**Position**				
Posterior Mandible	44	−1.90 (1.97)	39	−2.33 (2.47)
Other Positions	8	−3.16 (2.24)	5	−2.67 (2.18)
**Bone Quality**				
1	9	−1.39 (1.54)	8	−2.42 (2.72)
2	43	−2.24 (2.12)	36	−2.36 (2.39)
**Bone Quantity**				
A	22	−1.44 (1.72)	19	−2.40 (2.63)
B	25	−2.36 (2.12)	21	−2.25 (2.37)
C	3	−2.53 (2.36)	3	−1.92 (1.38)
D	2	−5.20 (0.99)	1	−5.65 (-)
**Platform**				
RP(Ø 4.3 mm)	25	−2.12 (1.94)	20	−1.80 (1.64)
WP(Ø 5.0 mm)	27	−2.07 (2.17)	24	−2.84 (2.86)
**Implant Length**				
10 mm	23	−1.73 (2.24)	19	−1.73 (2.25)
13 mm	26	−2.50 (1.88)	23	−2.95 (2.53)
16 mm	3	−1.35 (1.59)	2	−1.85 (2.05)
**Flap Design**				
No flap	3	−4.52 (1.50)	3	−3.18 (0.45)
Punch	5	−3.61 (2.78)	4	−2.67 (2.52)
Flap	44	−1.76 (1.83)	37	−2.27 (2.53)
**Site**				
Immediate	3	−2.65 (1.23)	3	−2.20 (1.42)
Healed	49	−2.06 (2.09)	41	−2.38 (2.49)
**Bone Grafting**				
No	23	−2.04 (2.08)	21	−1.82 (2.06)
Yes	29	−2.14 (2.05)	23	−2.87 (2.65)
**Insertion Torque**				
≤45	37	−2.15 (2.14)	29	−1.96 (2.09)
>45	15	−1.95 (1.83)	15	−3.17 (2.86)

SD, standard deviation; CI confidence interval. r, correlation coefficient.

## Data Availability

The datasets generated and analyzed during the current study are available from the corresponding author on reasonable request.

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
