# Peer review of "One-Piece Zirconia Oral Implants for the Support of Three-Unit Fixed Dental Prostheses: Three-Year Results from a Prospective Case Series"

_jfb, 2023, doi:10.3390/jfb14010045_

Round 1

Reviewer 1 Report

The authors presented in their manuscript One-piece zirconia oral implants for the support of three-unit fixed dental prostheses: Three-year results from a prospective case series valuable results of zirconia oral implants.

For good osteointegration of an implant, a rough surface is needed. For Ti6Al4V implants, corundum grit blasting is widely used. The disadvantage of this mechanical procedure is retaining corundum on the rough surface and also the sub-surface for up to 20%, these corundum particles caused  the aseptic loosening.

Please add the data about roughening procedure of  zirconia implant surface

Reviewer 2 Report

The paper "One-piece zirconia oral implants for the support of three-unit fixed dental prostheses: Three-year results from a prospective case series." is suitable for publication JFB Journal. The research study is interesting and has novlety data.

In introduction part (line 57), the authors should update the state-of-the-art reference with alternative coatings like APS. "As an alternative, there are additive methods to achieve porous surfaces 57 e.g. by coating the implants with a zirconia powder......"  Suggested reference : " In vitro electrochemical properties of biodegradable ZrO2-CaO coated MgCa alloy using atmospheric plasma spraying", Journal of Optoelectronics and Advanced Materials Volume 17, Issue 7-8, Pages 1186 - 11921 July 2015

The study should include some microstructural analysis (like OM, SEM or alternative)

Rest is fine.
